# The Functions of Chloroplastic Ascorbate in Vascular Plants and Algae

**DOI:** 10.3390/ijms24032537

**Published:** 2023-01-28

**Authors:** Szilvia Z. Tóth

**Affiliations:** Laboratory for Molecular Photobioenergetics, Institute of Plant Biology, Biological Research Centre, Temesvári krt 62, H-6726 Szeged, Hungary; toth.szilviazita@brc.hu

**Keywords:** ascorbate, non-photochemical quenching, oxygen-evolving complex, photosynthesis, photosystem II, reactive oxygen species, vitamin C

## Abstract

Ascorbate (Asc) is a multifunctional metabolite essential for various cellular processes in plants and animals. The best-known property of Asc is to scavenge reactive oxygen species (ROS), in a highly regulated manner. Besides being an effective antioxidant, Asc also acts as a chaperone for 2-oxoglutarate-dependent dioxygenases that are involved in the hormone metabolism of plants and the synthesis of various secondary metabolites. Asc also essential for the epigenetic regulation of gene expression, signaling and iron transport. Thus, Asc affects plant growth, development, and stress resistance via various mechanisms. In this review, the intricate relationship between Asc and photosynthesis in plants and algae is summarized in the following major points: (i) regulation of Asc biosynthesis by light, (ii) interaction between photosynthetic and mitochondrial electron transport in relation to Asc biosynthesis, (iii) Asc acting as an alternative electron donor of photosystem II, (iv) Asc inactivating the oxygen-evolving complex, (v) the role of Asc in non-photochemical quenching, and (vi) the role of Asc in ROS management in the chloroplast. The review also discusses differences in the regulation of Asc biosynthesis and the effects of Asc on photosynthesis in algae and vascular plants.

## 1. Introduction

Ascorbate (Asc) is the most abundant water-soluble metabolite in plants and is essential for plant growth and development. Asc is also required in the human diet; therefore, serious efforts are underway to increase the Asc contents of fruits and vegetables (reviewed, e.g., by [1,2]). In vascular plants and green algae, Asc is synthesized via the Smirnoff-Wheeler pathway, in which GDP-D-mannose is converted to L-galactono-1,4-lactone in the cytoplasm by GDP-D-mannose 3′,5′ epimerase (GME), GDP-L-galactose phosphorylase (GGP), L-galactose-1-P phosphatase (GPP), and L-galactose dehydrogenase (GDH) [3,4]. The final step, the conversion of L-galactono-1,4-lactone to Asc, occurs in the mitochondria and is catalyzed by L-galactono-1,4-lactone dehydrogenase (GLDH) at Complex I [4,5] (Figure 1). *VTC2*, encoding GGP, plays a vital role in the regulation of Asc biosynthesis, both in vascular plants and green algae [6,7]. In vascular plants, not only its expression but also a feedback mechanism on GGP translation by Asc and a small ORF largely determines the rate of Asc biosynthesis, thereby its cellular concentration [8,9,10,11]. 

Alternative Asc biosynthesis routes, namely, the galacturonate, the L-gulose pathway, and the myoinositol pathway, have been suggested to contribute to Asc biosynthesis (reviewed by [12]). However, various lines of evidence show that they do not contribute significantly to Asc biosynthesis in plant leaves [13]. 

The best-known role of Asc is to scavenge reactive oxygen species (ROS), in a highly regulated manner in each cellular compartment (reviewed by [14]). It is linked to its capacity to act as a weak reductant and to the non-toxicity of the generated monodehydroascorbate (MDA) and the bicyclic dehydroascorbate (DHA) [15,16]. Asc effectively eliminates superoxide and tocopherol radicals, though it reacts slowly with H_2_O_2_. Plants use Asc peroxidases (APX) to eliminate H_2_O_2_ more effectively (reviewed by [14,15]). The reaction oxidizes Asc into MDA radicals, which can spontaneously disproportionate into DHA and Asc. The MDA radicals are then recycled back to Asc through MDA reductases (MDARs) or reduced ferredoxin produced at the acceptor side of photosystem I (PSI) [14,17], while DHA is reduced by DHA reductase (DHAR) in glutathione-dependent non-enzymatic or enzymatic reactions [18,19] (Figure 1).

DHA is unstable; therefore, if it is not recycled back to Asc, it is rapidly and irreversibly degraded into oxalate, L-threonate, or tartaric acid, depending on the plant species [20,21]. Nevertheless, the regeneration of DHA and MDA occurs very effectively, as most of the Asc pool is in the reduced state under both non-stress and mild stress conditions. An exception in this respect is the apoplast in which the oxidation of Asc occurs at a relatively high rate due to the activity of Asc oxidase; this peculiarity is most probably related to cell wall loosening and growth [22,23]. 

Besides being an effective antioxidant, Asc also acts as a chaperone for 2-oxoglutarate-dependent dioxygenases (2-ODDs). Plant 2-ODDs are involved, for instance, in hormone metabolism (ethylene, abscisic acid, gibberellins, indole-3-acetic acid) and the synthesis of various secondary metabolites, such as anthocyanins and glucosinolates [24,25,26]. Asc is also involved in the epigenetic regulation of gene expression [27] via modifying the activities of ten-eleven translocation (TET) and Jumonji C-domain-containing histone demethylases that are both 2-ODDs [27,28,29,30]. Asc has also been proposed to be involved in iron transport and Ca^2+^ signaling [31,32]. 

Thus Asc affects plant growth, development, and stress resistance via various mechanisms that are challenging to dissect. Growth defects have been reported for the Asc-deficient *vtc2-1* Arabidopsis mutant containing about 20% Asc relative to the wild type, but later it turned out to be due at least partially to cryptic mutations [33]. The more recently identified T-DNA mutant *vtc2-4* did not show any alteration in the phenotype under optimal growth conditions in [33], but a mild phenotype was observed in [34]. Moreover, the approx. 80% reduction in Asc level observed in the *vtc2* mutants only moderately enhances high light sensitivity [35]; thus, this low level of Asc seems to be sufficient for the housekeeping functions under laboratory conditions. On the other hand, the situation can be different in the field, and it may also be species-dependent because in rice, for instance, Asc content reduction did lead to diminished growth and photosynthesis rate [36]. 

Several excellent reviews have already been published on the biosynthesis and the various functions of Asc (e.g., [11,14,15,37,38,39,40]). This review focuses on less extensively discussed issues, such as the relationship between photosynthesis and Asc biosynthesis and the effects of Asc on photosynthetic electron transport in vascular plants and algae.

## 2. Regulation of Ascorbate Biosynthesis by Light 

Under regular growth and moderate light conditions (at approx. 100 μmol m^−2^s^−1^), the Asc level of Arabidopsis leaves is in the range of 2 to 5 μmol/g fresh weight (FW). The Asc content varies by a factor of two depending on the time of day, with a minimum at night and a maximum in the afternoon [41,42,43]. Under stress conditions, including high light [9,42,44,45], ozone [46,47], salt [48,49], drought stress [50], and nitrogen starvation [51] a two- to three-fold increase in Asc level occurs on a timescale of days, with an apparent upper limit of about 8–10 µmol Asc/g FW. Using stable and inducible expression systems in Arabidopsis, the maximum Asc concentration that can be reached is in the same range of maximum 10 µmol Asc/g FW (e.g., [7]; reviewed by [11,38]). Asc content can be somewhat more effectively increased by L-galactono-1,4-lactone treatment to the range of 15 µmol Asc/g FW [52,53].

These results demonstrate that Asc biosynthesis is highly regulated, to keep Asc in the optimum concentration range during the normal life cycle of plants and under stress conditions. Therefore, understanding the signaling pathways and the control mechanism of Asc biosynthesis are of primary importance to achieve a substantial increase in the Asc content of crops. 

One major factor controlling Asc biosynthesis is light. Several Smirnoff-Wheeler pathway genes are induced by illumination. These include GDP-D-mannose pyrophosphorylase (GMP), GGP, GPP, and GLDH [54,55,56]. Transcript levels of these biosynthesis genes also follow a circadian rhythm [42]. It has been shown that 3-(3,4-dichlorophenyl)-1,1-dimethylurea (DCMU), an inhibitor of photosystem II (PSII), prevents the increase in Asc pool size in Arabidopsis upon a shift from light-dark cycle to continuous light conditions, along with a decrease in the transcript levels of GMP, GGP, GPP, and GLDH [55]. 

When Arabidopsis plants were transferred to a medium containing sucrose, the leaf Asc levels decreased along with a decrease in the rate of CO_2_ fixation. Asc content diminishment upon sucrose feeding did not occur in a sugar-insensitive Arabidopsis *abi4/sun6* mutant [55]. Exogenous application of glucose in pea seedlings did not affect Asc content [57], whereas sucrose feeding in detached broccoli inflorescences delayed Asc depletion [58], and in tomato fruit, it increased the Asc content [59]. 

The Asc content decreases markedly in darkness (at a rate of approx. 2% per hour), while the levels of its degradation products increase [21], and dark-grown plants produce no Asc at all [60]. Interestingly, the downregulation of *VTC2* expression by DCMU was slightly reversed in Arabidopsis mutants lacking GENOMES UNCOUPLED 1 (GUN1); [61], the master regulator of chloroplast-to-nucleus retrograde signaling. 

Thus, the results suggest that photosynthesis-derived signal(s) participate in the light activation of Asc biosynthesis, and the role of photosynthesis is not solely to provide a carbon source for Asc biosynthesis. On the other hand, soluble carbohydrates may affect the expression of Asc biosynthesis genes and the regeneration of Asc. The molecular mechanisms underlying the regulation of Asc biosynthesis by photosynthesis remain to be explored. It is conceivable that the redox state of the plastoquinone pool plays a regulatory role in Asc biosynthesis. In principle, H_2_O_2_ and other ROS produced during photosynthesis could also influence the expression of Asc biosynthesis genes; however, experimental evidence is still lacking on such roles.

Light may also directly affect Asc biosynthesis, degradation, and regeneration. Indeed, light can regulate GMP activity, namely, via CSN5B, a photomorphogenic factor that is part of a CSN complex, negatively regulating photomorphogenesis in Arabidopsis via proteasomal degradation [62,63,64]. Wang et al. [64] demonstrated that GMP is polyubiquitinated and degraded in darkness via interaction with CSN5B. They also found that loss of CSN5B function impaired the effect of light on Asc synthesis in response to continuous light or darkness, showing that CSN5B is a posttranslational regulator in Asc biosynthesis. 

A F-box type repressor, known as Asc acid mannose pathway regulator 1 (AMR1), also regulates Asc biosynthesis in a light-dependent manner [65]. DNA knockout lines for AMR1 accumulated two-fold greater foliar Asc than the wild type. AMR1 also negatively affected the expression levels of most Asc biosynthesis genes, including GMP, GME, GGP, GPP, L-GalDH, and GLDH. In addition, AMR1 expression was higher in aging leaves, and lower at medium light than at low light intensity. 

A HD-Zip I family transcription factor in tomato, SlHZ24, positively regulates the accumulation of Asc by binding to the promoter of SlGMP3, and it also regulates the expression of GME and GGP. Accordingly, the Asc content fluctuated following the expression of SlHZ24 in a light-dependent manner [66].

GGP also shows a significant response to different light levels due to the regulation by light-responsive cis-elements in its promoter, namely, a G-box motif [67]. Furthermore, light-responsive cis-elements have also been identified in the promoters of GPP and GLDH in rice [68]. 

It was also proposed that the VTC3 dual protein kinase/protein phosphatase is involved in signal transduction to adjust Asc levels in response to light and temperature changes [69]. VTC3 is probably located in the chloroplast, and it was suggested that the control exerted by VTC3 is post-transcriptional and does not alter the transcript levels of Asc biosynthesis genes in Arabidopsis [69].

The conversion of L-galactono-1,4-lactone to Asc in the mitochondria is light and photosynthesis dependent because photosynthetic inhibitors prevent it [52]. H_2_O_2_ produced upon stress effects may inactivate selectively and reversibly Arabidopsis GLDH by oxidizing a cysteine residue (Cys-340) [70]. GLDH is protected from inactivation both by L-galactono-1,4-lactone and Asc; therefore, their availabilities and the level of H_2_O_2_ may affect the rate of Asc biosynthesis in vascular plants. 

Bryophytes, including *Brachytecium velutinum*, *Marchantia polymorpha*, and *Physcomitrium* (formerly *Physcomitrella*) *patens*, possess Asc content slightly less than seed plants, in the range of 0.3 to 2 μmol/g [71,72]. Sodeyama et al. [72] presented evidence on the Smirnoff-Wheeler pathway as a source of Asc, whereas the D-galacturonate pathway did not seem to contribute to Asc biosynthesis in *P. patens*. Interestingly, two *VTC2* paralogs are functional in *P. patens*, and, in contrast to higher plants, they are both equally responsible for Asc biosynthesis. Furthermore, the light-induced fluctuation of the transcript level of the two *VTC2* genes was comparable to *AtVTC2*. Additionally, DCMU treatment diminished *VTC2* expression and Asc content, as observed earlier in Arabidopsis. Thus, *VTC2* expression is a crucial control point of Asc biosynthesis in *P. patens*. Interestingly, knockout mutants with low Asc content exhibited restricted side branch growth in their protonemata; this may be caused by multiple factors related to the various physiological functions of Asc.

Green alga grown under benign conditions contain about 100-fold less Asc than vascular plants, i.e., in the range of 100 to 500 μM [73,74,75]. Asc is synthesized via the Smirnoff-Wheeler pathway in green algae; however, its regulation significantly differs compared to plants. It has been shown that in *Chlamydomonas reinhardtii*, the expression of the *VTC2* gene is induced by H_2_O_2_ and ^1^O_2_, resulting in a strong increase in Asc content. On the other hand, photosynthesis is not directly required for Asc biosynthesis. Additionally, in contrast to plants, there is no circadian regulation of Asc biosynthesis, and *C. reinhardtii* lacks negative feedback regulation by Asc in the physiological concentration range. These mechanisms enable a rapid and manifold increase in Asc content upon various stress treatments, including light stress and sulfur deprivation [74,76,77].

## 3. Interaction between Photosynthetic and Mitochondrial Electron Transport in Relation to Asc Biosynthesis

Photosynthetic electron transport rates (ETR) and NADPH levels are only slightly affected in *vtc2* mutants containing about 20% Asc in comparison with wild-type Arabidopsis plants when grown under moderate or low light conditions [35,44,78,79], whereas the photosynthetic ETR of *vtc2* mutants is diminished at high light [44]. The *vtc2* mutants have a slightly lower stomatal conductance, which may be related to a regulatory effect of Asc [79,80]; this, however, is compensated by a larger stomatal number and increased RuBisCO content. Therefore, the overall CO_2_ assimilation rate is affected by Asc deficiency only in high light, but not under normal growth conditions in Arabidopsis [79,81].

The last step of Asc biosynthesis, the conversion of L-galactono-lactone into Asc, is catalyzed by GLDH in the mitochondria [82,83]. GLDH is a protein of 58 kDa, located at Complex I in the mitochondrial intermembrane space, tightly tethered to the membrane through protein-protein interactions [84] (Figure 1). Complex I is organized in two arms: the matrix arm transfers electrons from NADH to ubiquinone, and the membrane arm is responsible for proton translocation [85]. In addition to being essential for Asc biosynthesis, GLDH has a non-enzymatic role in the assembly of the membrane arm of Complex I [86]. This function is likely to be independent of the role of GLDH in Asc biosynthesis because the *vtc2-1* Arabidopsis mutant accumulates wild-type levels of Complex I [86].

During the oxidation of L-galactono-1,4-lactone to Asc by GLDH, electrons are fed into the mitochondrial electron transport chain via cytochrome c [5,87], a soluble redox-active heme protein that transfers electrons from Complex III to Complex IV [88] (Figure 1). Cytochrome c knockdown mutants of Arabidopsis had a 60% decrease in GLDH activity without affecting the Asc content, showing that low cytochrome c levels are enough under normal growth conditions to sustain Asc biosynthesis [89]. 

Plant mitochondria also have an alternative oxidase (AOX) pathway, taking electrons directly from the ubiquinone pool without the contribution of the cytochrome c pathway (Figure 1). Bartoli et al. [90] found that AOX-overexpressing Arabidopsis lines accumulated more Asc than wild-type plants, particularly at high light. Higher throughput in the cytochrome c pathway would require a larger pool of electron acceptors for L-galactono-1,4-lactone oxidation; therefore, an enhanced capacity of the AOX pathway may favor Asc biosynthesis by maintaining the cytochrome c pool in a more oxidized state. This is particularly relevant under high light conditions to prevent over-reduction of the mitochondrial electron transport chain and, at the same time, to enhance Asc biosynthesis to protect the cells against the damaging effect of ROS [90]. These results show that integration of L-galactono-1,4-lactone oxidation and mitochondrial electron transport chain activity via cytochrome c could coordinate Asc biosynthesis and respiration [88]. 

It has also been shown that the conversion of L-galactono-1,4-lactone depends on the photosynthetic electron transport chain because DCMU and dibromothymoquinone could effectively inhibit the Asc content increase upon L-galactono-1,4-lactone treatment [52]. This result indicates the occurrence of a crosstalk between photosynthetic and respiratory electron transport chains (Figure 1). 

It has been suggested that Asc may be a signal connecting the metabolisms of chloroplast and mitochondria [37,91] based on observations on transgenic tomato plants antisensed in mitochondrial malate dehydrogenase (*mdh*). When grown under long-day conditions, these *mdh* lines had reduced tricarboxylic acid (TCA) cycle activity without affecting respiration, and intriguingly, CO_2_ assimilation rates and carbohydrates were slightly enhanced compared to wild-type plants, and an approx. fourfold increase in Asc content occurred [92]. This was explained by an upregulated flux through GLDH in the *mdh* lines and a higher capacity to use L-galactono-lactone as a respiratory substrate, thereby suggesting that GLDH can effectively act as an alternative electron donor in cases where flux through the TCA cycle is impaired [91,92]. However, contradicting results were obtained under short-day conditions [91] and in Arabidopsis *mdh* mutants [93]. On the other hand, Asc feeding to isolated leaf discs also resulted in increased photosynthesis rates, further suggesting an Asc-mediated link between the energy-generating processes of respiration and photosynthesis.

In summary, these results suggest that the interaction between chloroplasts and mitochondria acts as a vital determinant of the light-dependent regulation of Asc biosynthesis in plants and that Asc may act as a metabolic regulator between the energy systems of the mitochondria and chloroplasts [37]. However, the mechanistic details and the sensing system remain to be explored.

## 4. Ascorbate Is an Alternative, ‘Emergency’ Donor to Photosystem II

Asc is a weak reductant that has the potential to reduce amino acid radicals, such as tyrosine and tryptophan [94]. This feature makes it capable of donating electrons to Tyr_Z_^+^ in PSII with inactive oxygen-evolving complexes (OEC). This was initially demonstrated in vitro on TRIS-washed, UV-B-irradiated, and heat-treated isolated thylakoids [95,96,97] and later on heat-treated intact leaves [98,99].

Heat stress results in the removal of the extrinsic proteins and the release of Ca- and Mn-ions from their binding sites, resulting in the inactivation of OEC [100,101]. It has been shown that electron donation from Asc to Tyr_Z_^+^ occurs in heat-stressed leaves; thus, Asc is a naturally occurring electron donor that can replace water, the terminal electron donor of PSII [99] (Figure 1). With the aid of chlorophyll *a* fluorescence induced by short (5-ms) light pulses it was shown that the halftime of electron donation from Asc to PSII is in the range of 25 ms in wild-type Arabidopsis leaves and about 55 ms in Asc-deficient *vtc2* mutants [78,99]. This alternative electron transport occurs in Arabidopsis, pea, barley, *Marchantia polymorpha*, *Nephrolepis exaltata*, *C. reinhardtii*, etc.; thus, it appears to be ubiquitous in the plant kingdom [99]. The electron transfer rate from Asc to PSII depends on the species and their physiological state, which is most probably related to the availability of Asc in the lumen (probably in the range of a few mM, [102]). 

Isolated photosynthetic samples with inactivated OECs are extremely susceptible to illumination. The impaired electron donation from the OEC results in the accumulation of highly oxidizing radicals, including P680^+^, Tyr_Z_^+^, and superoxide and hydroxyl radicals [103,104], leading to a rapid inactivation and degradation of PSII reaction centers [105,106]. This type of photodamage is called weak light or donor-side-induced photoinhibition. By using intact leaves of wild-type, Asc-overproducing (*miox4*) [107], and Asc-deficient Arabidopsis mutants (*vtc2-3*) [6] subjected to heat stress (40 °C, 15 min), it was demonstrated that the continuous electron flow from Asc to PSII alleviates PSII photoinactivation [78]. Gradual inactivation of PSII charge separation activity occurred on a time scale of tens of minutes, along with extensive protein degradation, including probably the complete disassembly of PSII [78]. Besides the rate of photoinactivation, the recovery rate from the photoinactivated state also depended on leaf Asc content. Thereby, Asc contributes significantly to the ability of plants to withstand heat stress conditions and aids recovery [78,108].

Asc may also act as an alternative electron donor in bundle sheath chloroplasts. These are found in the so-called NADP^+^ malic enzyme type species carrying out C4-photosynthesis, such as maize and sorghum. The amount of PSII in bundle sheath chloroplasts is small, and their OECs have low activity. It was shown that Asc is an effective electron donor for PSII in bundle sheath chloroplasts in vivo [109,110]. On the other hand, since the number of PSII reaction centers is low, photosynthetic electron transport is moderate, but it is sufficient to maintain PSI cyclic electron flow, ensuring thylakoid membrane energization and ATP synthesis for the Calvin-Benson cycle [109,110]. Moreover, the replacement of water by Asc as a PSII electron donor also ensures low O_2_ concentration within the bundle sheath chloroplasts, diminishing the risk of competition of O_2_ with CO_2_ molecules for the catalytic sites of RuBisCO (reviewed by [111]). 

Asc also donates electrons to PSI in isolated thylakoid membranes (see, e.g., [112]) and in DCMU-treated bundle sheath cells isolated from maize leaves [113]. However, in vivo, Asc was a far more effective electron donor for PSII than for PSI [99,109,110].

## 5. Ascorbate May Impair the Oxygen-Evolving Complex

When considering the physiological roles of Asc, it has to be taken into account that it is a reductant, and therefore, its cellular concentration is to be maintained in a particular range [114]. The basal Asc concentration in green algae is very low compared to higher plants (approx. 60 to 100 µM in *C. reinhardtii* and 5 mM in Arabidopsis) [74,75,77,115]. It was observed that upon sulfur deprivation of *C. reinhardtii*, Asc accumulates to the mM range and that, in this range, Asc over-reduces the Mn cluster of OEC [76,77]. The exact mode of action, i.e., as to which S-state is being inactivated, is not understood.

Once the Mn-cluster is over-reduced by Asc, it may continuously provide electrons to Tyr_Z_^+^. However, the electron donation by Asc to PSII is relatively slow (halftime of approx. 20 to 50 ms, [99]) in comparison with the rate of electron transfer from intact OECs to Tyr_Z_^+^ (halftime of about 0.1 to 1 ms; [116]); for this reason, Asc cannot entirely prevent the accumulation of Tyr_Z_^+^ and P680^+^ upon illumination in sulfur-deprived *C. reinhardtii* cultures. Furthermore, strongly oxidizing species lead to donor-side induced photoinhibition, resulting in the relatively rapid degradation of PSII reaction center proteins, including PsbA, CP43, PSBO, and possibly others [77]. 

The loss of PSII activity could be regarded as damage induced by sulfur deprivation. However, upon downregulation of PSII activity, overexcitation and further photodamage are minimized. The metabolic changes downregulating photosynthetic activity and cell proliferation may serve to preserve cellular sulfur content and avoid more substantial damage [77]. On the other hand, the inactivation of OECs contributes to the establishment of hypoxia enabling hydrogenase expression, and H_2_ production will act as a safety valve for photosynthetic electron transport [117]. Thereby, the damage imposed by sulfur limitation is minimized, and the alga cells may recover if sulfur becomes available again [118].

Intriguingly, Asc inactivates the OEC in green algae when accumulated to the mM range, but the same Asc concentration in vascular plants is physiological [115], which enables a full operation of OEC activity. On the other hand, it was demonstrated earlier that upon the chemical removal of the extrinsic OEC subunits in isolated PSII membranes, bulky reductants, including Asc, could directly reduce the Mn-cluster [119]. 

There are notable differences between the extrinsic OEC proteins of vascular plants and green algae [120,121]. Namely, in green algae, the Mn-cluster is shielded by a 33 kDa PSBO subunit and two smaller subunits, PSBP and PSBQ, with some structural differences and binding properties in comparison to vascular plants [121,122]. Moreover, in vascular plants, an additional subunit, PSBR, is found in the vicinity of the Mn-cluster [123]. The structural differences and the variance in binding properties of the extrinsic proteins may be key in explaining why Asc reduces the Mn-cluster in algae but not in higher plants when present in the same concentration range. It is also conceivable that, during evolution, as cellular Asc concentration increased [124], the extrinsic proteins have evolved to protect the OEC against the reducing effect of Asc present in the thylakoid lumen. 

In Arabidopsis subjected to darkness for 24 h, the Mn-cluster becomes inactivated, being one of the earliest effects of dark-induced senescence on photosynthesis [125]. Remarkably, the extent of OEC inactivation was much weaker in Asc-deficient mutants compared to wild-type plants, suggesting that Asc was responsible for the diminishment of oxygen evolution. In a *psbo1* knockout mutant, the compromised OEC activity was further aggravated upon dark treatment, suggesting that the extrinsic proteins protect the OEC against the reducing effect of Asc. In the absence of PSBR, only a slightly disturbed photosynthetic activity was observed under normal growth conditions, whereas a strongly diminished OEC activity was observed in the dark. A double *psbo1 vtc2* mutant showed a slightly milder photosynthetic phenotype than that of the single *psbo1* mutant [125]. Thus, these results suggest that Asc over-reduces the Mn-complex in prolonged darkness. This is probably enabled by a dark-induced dissociation of the extrinsic OEC subunits, which would otherwise hinder the access of Asc to the Mn-cluster [125] (Figure 1) or, hypothetically, it may be related to the lumen volume decrease in the dark [126].

Another example that Asc may negatively affect certain cellular processes was found by Castro et al. [127]. Exogenous Asc concentrations above 30 mM caused cellular and oxidative damage that were enhanced by high light. The high Asc concentration induced H_2_O_2_ accumulation, stomatal closure, and impairment in CO_2_ assimilation, as well as photosynthetic electron transport. Therefore, these data show that Asc concentration and localization need to be highly controlled, particularly relevant when aiming at generating crops with elevated Asc contents.

## 6. The Role of Ascorbate in Non-Photochemical Quenching

Excess light may lead to photooxidative stress involving the formation of ROS and light damage (recently reviewed by, e.g., [128,129,130]). Non-photochemical quenching (NPQ) of excitation energy is a complex photoprotection mechanism, including energy-dependent (qE), zeaxanthin-related (qZ), state-transition-related (qT), and photoinhibitory (qI) components (e.g., [131,132]). 

The pH-regulated qE component is formed in response to light-intensity changes within a few minutes. In vascular plants, activation of qE is mediated by PsbS that acts as a lumen pH sensor and induces LHCII antenna rearrangement in a zeaxanthin-dependent manner [133,134]. 

The qZ component of NPQ is activated in the time range of 10 to 30 min, which correlates with the formation of zeaxanthin by violaxanthin de-epoxidase (VDE) in the lipid phase of the thylakoid membrane of vascular plants [135,136,137]. VDE belongs to the lipocalin protein family, and it utilizes Asc as a co-substrate (Figure 1), providing the reducing power for de-epoxidation [138]. At the pH-optimum of VDE of pH 5.0, the Km value for Asc is approximately 1.0 mM [138,139], which is probably in the range of luminal Asc concentration [102]. During the operation of the violaxanthin cycle, VDE attaches to the luminal side of the thylakoid membrane following its pH-dependent activation [140,141,142]. The active VDE is probably a dimer, capable of binding violaxanthin and Asc [141,143].

The maximum VDE activity is several times faster than zeaxanthin epoxidase activity [144]. Consequently, zeaxanthin accumulates in strong light when a low lumen pH is established. In contrast, zeaxanthin is epoxidized only when VDE activity is low, i.e., under low light or in darkness. It was recently demonstrated that up-regulation of VDE, PsbS, and zeaxanthin epoxidase in soybean significantly accelerated the violaxanthin cycle, leading to faster induction and relaxation of NPQ. This increased the efficiency of CO_2_ assimilation and PSII electron transport in fluctuating light conditions and significantly improved the biomass yield in field studies, demonstrating that the violaxanthin cycle plays a crucial role in the regulation of photosynthesis, thereby relating to plant productivity [145].

Besides playing a role in NPQ, zeaxanthin may also contribute to photoprotection by acting as an antioxidant and possibly by modulating thylakoid membrane properties [146,147]. In addition, zeaxanthin may be required for the PSII repair cycle [148]. 

The physiological importance of Asc for zeaxanthin formation was demonstrated in vivo using *vtc* mutants of Arabidopsis: it was shown that they accumulate less zeaxanthin in high light, and consequently, they have diminished and/or delayed NPQ induction [6,45,78]. In the *miox4* Asc-overproducing mutant [107], a slightly higher NPQ level was obtained than in the wild type [78], demonstrating that Asc may be limiting NPQ formation [44].

In contrast to the *vtc2* mutant of Arabidopsis, the *C. reinhardtii vtc2* mutant (*Crvtc2-1*) with strongly decreased Asc content performs normal violaxanthin de-epoxidation [75]. *C. reinhardtii* lacks a plant-type VDE and instead uses an unrelated enzyme belonging to lycopene cyclases, called *Chlorophycean* VDE (CVDE). It is located on the stromal side of the thylakoid membrane [149], and it does not require Asc for violaxanthin de-epoxidation [75]. On the other hand, a slow, H_2_O_2_-dependent NPQ component (probably qI) is enhanced upon Asc-deficiency in *C. reinhardtii* [75]. 

Many green alga species, including *Chlorella vulgaris*, contain plant-type VDEs [150], which are crucial for photoprotective NPQ. Thus, there is an evolutionary divergence of photoprotective mechanisms among *Chlorophyta* [151]. *Chromalveolate* algae, such as *Phaeodactylum tricornutum*, use the diadinoxanthin cycle instead of the violaxanthin cycle in a fashion similar to vascular plants to support qE [152]. PtVDE converts the monoepoxide diadinoxanthin to diatoxanthin, possibly involving violaxanthin as an intermediate [153]. The reaction requires Asc, though less than vascular plants [154]. On the other hand, violaxanthin is also required for the biosynthesis of fucoxanthin and its derivatives, which are the main light-harvesting pigments in *Chromalveolate* algae. The conversion of violaxanthin to neoxanthin is catalyzed by the so-called violaxanthin de-epoxidase-like (VDL) protein, which does not require Asc as a reductant. It was found that PtVDL is modulated only by pH, whereas VDE activity is controlled on multiple levels, including the pH-dependent affinity for Asc [155].

Mosses have plant-type VDE enzymes [156], which probably require Asc as a reductant. The regulation of Asc biosynthesis in vascular plants and mosses is similar [72], but the Asc-dependence of NPQ in mosses has not been investigated.

## 7. Reactive Oxygen Species Management by Ascorbate in the Chloroplast

The best-known and most extensively discussed role of Asc is participation in ROS management. Excellent recent reviews are available on this topic [14,129,130]; therefore, only a few aspects related to photosynthesis are mentioned here.

Asc is essential for the enzymatic scavenging of ROS in the so-called Mehler reaction or the water–water cycle (reviewed by e.g., [14,157], Figure 1). This reaction becomes particularly relevant at high light intensities and/or when the Calvin-Benson cycle cannot work at high speed, for instance under drought, cold, or salt stress. Under such conditions, the outflow of electrons at the PSI acceptor side is inhibited, leading to a surplus of electrons; therefore, ferredoxin reduces O_2_, and superoxide is produced. It is then reduced to H_2_O_2_ by superoxide dismutase, which is reduced by APX to water. MDA can be directly reduced back to Asc by PSI, and/or in the Asc-glutathione cycle in which MDAR and DHAR use NADPH as reducing power [158]. Asc thereby participates in the mitigation of ROS. In this respect, it is to be considered that ROSs are key signaling molecules that enable cells to respond to changes in environmental conditions rapidly, and they integrate different environmental signals to activate stress-response networks and defense mechanisms [130]. 

In addition to its function as a powerful antioxidant, Asc is also considered as being a major player in redox homeostasis and part of the redox signaling network that regulates plant responses to biotic stress [14,159,160]. Interestingly, however, an approx. 80% decrease in Asc content led to only a minor increase in glutathione level when plants were grown under standard laboratory conditions [35,81], though cellular glutathione levels are redistributed and an approx. two-fold increase in chloroplastic glutathione concentration was observed [161]. In addition, Asc peroxidase activity and the redox state of Asc are also unaltered in *vtc2* mutants, just as well as the photosynthetic activity and carotenoid content, and only moderate changes occur at constant high light in these parameters [33,35]. Thus, an approx. 80% reduction of cellular Asc content does not lead to photooxidative stress, and apparently, Asc is in large excess in vascular plants, or its deficiency, are compensated via various yet unexplored mechanisms.

## 8. Open Questions and Perspectives

Future work will be required to gain a complete overview of Asc functions within the cell. In recent years, several functions of Asc have been discovered in addition to its best-known role of mitigating ROS accumulation, and it is likely that the list will be further expanded. For instance, Asc serves as a chaperone for 2-ODD, a versatile oxidative enzyme group in plants, of which only a couple have been characterized [162], and it is conceivable that Asc regulates the activities of some of them. In addition, the role of Asc as a metabolic regulator and a key player in chloroplast-mitochondria signaling has been suggested [37], which warrants experimental confirmation. Moreover, Asc transporters may also play a key role in Asc metabolism and functions of which only two have been characterized on a molecular level in vascular plants (AtPHT4;4 and AtDTX25 in the chloroplast envelope membrane and the vacuole, respectively [163,164]). Therefore, future work needs to be directed at exploring other essential Asc transporters both in vascular plants and algae.

## Figures and Tables

**Figure 1 ijms-24-02537-f001:**
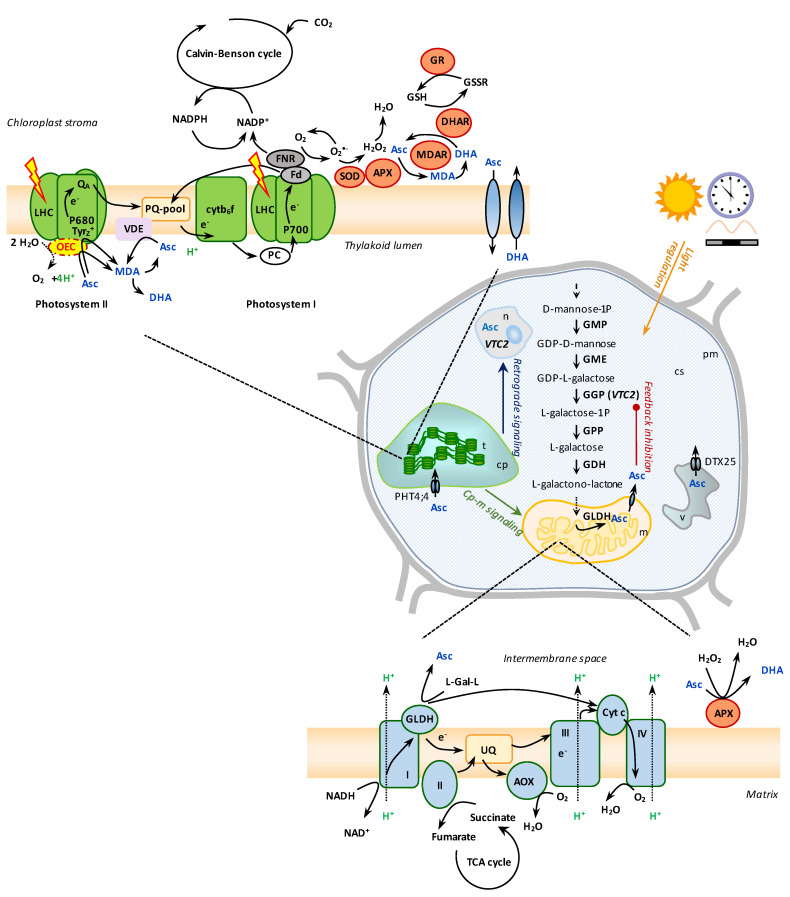
Ascorbate (Asc) biosynthesis and the roles of chloroplastic Asc in vascular plants. Asc is synthesized via the Smirnoff-Wheeler pathway with the majority of the steps taking place in the cytosol and the last step in the mitochondria at Complex I. Asc biosynthesis is regulated by light, circadian clock, photosynthetic electron transport, and a feedback inhibition by Asc. Asc is found in all cell compartments; PHT4;4 transports it into the chloroplast. Chloroplastic Asc plays multiple roles: (i) it is an alternative electron donor of photosystem II (PSII) when the oxygen-evolving complex (OEC) is inactive, (ii) it may inactivate the OEC under specific circumstances, (iii) it is a co-substrate of violaxanthin de-epoxidase (VDE) thereby plays a role in non-photochemical quenching, (iv) Asc participates in reactive oxygen species management. Abbreviations: APX, Asc peroxidase; cp, chloroplast; cs, cytosol; Cytb_6_f, cytochrome b_6_f; Cytc, cytochrome c; DHA, dehydroascorbate; DHAR, dehydroascorbate reductase; DTX25, vacuolar ascorbate transporter; Fd, ferredoxin; FNR, ferredoxin-NADP oxidoreductase; GDH, L-galactose dehydrogenase; GGP, GDP-L-galactose phosphorylase (encoded by *VTC2*); GLDH, L-galactono-1,4 lactone dehydrogenase; GME, GDP-D-mannose 3′,5′-epimerase; GMP, GDP-D-mannose phosphorylase; GPP, L-galactose-1-P phosphatase; GR, glutathione reductase; L-Gal-L, L-galactono-lactone; LHC, Light-harvesting complex; m, mitochondrion; MDA, monodehydroascorbate; MDAR Monodehydroascorbate reductase; n, nucleus; PHT4;4, chloroplastic Asc transporter; pm, plasma membrane; PSI, photosystem I; SOD, superoxide dismutase; t, thylakoid; TCA, tricarboxylic acid; UQ, ubiquitine; v, vacuole.

## Data Availability

All data presented in this study are available within this article. There are no special databases associated with this manuscript.

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
