# Peer review of "The Functions of Chloroplastic Ascorbate in Vascular Plants and Algae"

_ijms, 2023, doi:10.3390/ijms24032537_

Round 1
Reviewer 1 Report
The manuscript by Tóth is a review describing the multiple roles of ascorbate in plants and algae cells. The author describes known, as well as only suggested ways of ascorbate involvement in photosynthetic apparatus functioning in general and in photosystem II (PSII) operation in particular.
The high scientific level of the author is felt, and, of course, I should recommend the manuscript for publication, but I would like to get the answers on my comments, as well as to give some recommendations to the author with aim to make the level of the review higher. Thus, I choose the ‘Major review’ only this time.
The title of second point (L85) has a mention about the photosynthetic electron transport chain (ETC). However, the major part of text below is devoted to the ’Regulation of ascorbate biosynthesis by light’. Thus, in my opinion, the author should cut the title or add more information about the role of ETC in the ’Regulation of ascorbate biosynthesis...’.
Only as a recommendation, I would like to see schemes of the thylakoid and mitochondrial membranes with lager size, while the scheme of the cell in contrast smaller in Figure 1. This is because the author below describes more the processes taking part in these membranes rather than in the whole cell.
I feel that two first sentences in the paragraph between L133 and L138 require references.
In L284 the author use and abbreviation OEC but it was not described above.
From the point four of the manuscript the author often use ‘we’ (L284, 288, 302, 306 etc.), but this a review and I think the author should avoid it in the entire text.
The author should indicate the species of the plants upon description of the results. For example - L258 ‘from Asc to TyrZ+ occurs in heat-stressed leaves’. The same L301 and through the entire text.
I think this is more correct to write ‘Once the Mn-cluster is overreduced by Asc’ (L335).
The author should use the same format for the PSII proteins indication, thus PsbA, PsbO, PsbP, PsbQ (L342, 360). At the same time, PsrP→PsbP (L360).
I absolutely disagree with the authors about the ‘remarkable differences between the extrinsic OEC proteins of vascular plants and green algae, (L358-359). The proteins indeed have some structural differences between plants and algae, but not so significant to easily explain the inactivation of OEC by ascorbate in green algae. At the same time, the author cites very suitable work to show the differences in association properties of the proteins in the case of plants and algae. I just recommend to the author to rewrite this paragraph and not to write so affirmatively here about structure differences.
I think that the first sentence in the paragraph from 369 requires references.
In L381 the author writes about the ‘a dark-induced dissociation of the extrinsic OEC subunits which would otherwise hinder the access of Asc to the Mn-cluster’. Probably it can be true, but the decrease in the lumen volume under dark also has to be taken into account. For example, this is a publication by Kirchhoff - DOI: 10.1073/pnas.1104141109. Thus, under dark the concentration of ascorbate in the lumen can increase and induce a stronger effect on the Mn-claster.
L392. ROS was abbreviated on first page, but NPQ was not described at all, as well as VAZ (L417).
In my opinion, the part of the point 6 from L391 to 439 is the weakest in the manuscript. This is written very confusingly with many abbreviations and in total with a little concern to ascorbate involvement. In addition, the author writes about PsbS, but not about Lhcsr3, which is an analogue in green algae. I would recommend to rewrite this part with a strong shortening to one paragraph.
In general it seemed to me that the manuscript describes all ascorbate presented in a plant (algae) cell but the title indicate ‘chloroplastic ascorbate’. Can the author explain to me this discrepancy? Should the title be corrected?
Author Response
Thank you very much for the positive evaluation of my MS. I have carefully considered and responded all the questions and criticisms.
"The manuscript by Tóth is a review describing the multiple roles of ascorbate in plants and algae cells. The author describes known, as well as only suggested ways of ascorbate involvement in photosynthetic apparatus functioning in general and in photosystem II (PSII) operation in particular.
The high scientific level of the author is felt, and, of course, I should recommend the manuscript for publication, but I would like to get the answers on my comments, as well as to give some recommendations to the author with aim to make the level of the review higher. Thus, I choose the ‘Major review’ only this time.
The title of second point (L85) has a mention about the photosynthetic electron transport chain (ETC). However, the major part of text below is devoted to the ’Regulation of ascorbate biosynthesis by light’. Thus, in my opinion, the author should cut the title or add more information about the role of ETC in the ’Regulation of ascorbate biosynthesis...’."
Response: I have shortened the title to ‘Regulation of ascorbate biosynthesis by light’.
"Only as a recommendation, I would like to see schemes of the thylakoid and mitochondrial membranes with lager size, while the scheme of the cell in contrast smaller in Figure 1. This is because the author below describes more the processes taking part in these membranes rather than in the whole cell. "
Response: I have modified the figure as suggested by the Reviewer.
"I feel that two first sentences in the paragraph between L133 and L138 require references."
Response: The relevant reference was added (Yabuta et al., 2007).
"In L284 the author use and abbreviation OEC but it was not described above."
Response: Amended.
"From the point four of the manuscript the author often use ‘we’ (L284, 288, 302, 306 etc.), but this a review and I think the author should avoid it in the entire text."
Response: To follow the recommendation of the Reviewer, I used passive voice instead of ‘we’.
"The author should indicate the species of the plants upon description of the results. For example - L258 ‘from Asc to TyrZ+ occurs in heat-stressed leaves’. The same L301 and through the entire text."
Response: A major part of the studies were carried out in Arabidopsis. However, we showed that the electron donation from Asc to PSII occurs in barley, pea, Marchantia, etc, as well. I have clarified this in the entire text.
"I think this is more correct to write ‘Once the Mn-cluster is overreduced by Asc’ (L335)."
Response: Amended.
"The author should use the same format for the PSII proteins indication, thus PsbA, PsbO, PsbP, PsbQ (L342, 360). At the same time, PsrP→PsbP (L360)."
Response: PSBO and the other extrinsic OEC subunits are nuclear-encoded therefore they should be written in full capitals. PsbA and PsaA are chloroplast-encoded proteins, therefore, only the first letter in the abbreviation (i.e. Psb for photosystem II) and the letter for the particular subunit should be written in capital. This way of spelling was employed in our most recent paper published in Plant, Cell and Environment (https://onlinelibrary.wiley.com/doi/epdf/10.1111/pce.14481) and in an earlier paper published in Plant Physiology (https://academic.oup.com/plphys/article/182/1/597/6116277).
Thank you for spotting the error; it should be PSBP instead of PSRP.
"I absolutely disagree with the authors about the ‘remarkable differences between the extrinsic OEC proteins of vascular plants and green algae, (L358-359). The proteins indeed have some structural differences between plants and algae, but not so significant to easily explain the inactivation of OEC by ascorbate in green algae. At the same time, the author cites very suitable work to show the differences in association properties of the proteins in the case of plants and algae. I just recommend to the author to rewrite this paragraph and not to write so affirmatively here about structure differences."
Response: I have softened the statements.
"I think that the first sentence in the paragraph from 369 requires references."
Response: Amended.
"In L381 the author writes about the ‘a dark-induced dissociation of the extrinsic OEC subunits which would otherwise hinder the access of Asc to the Mn-cluster’. Probably it can be true, but the decrease in the lumen volume under dark also has to be taken into account. For example, this is a publication by Kirchhoff - DOI: 10.1073/pnas.1104141109. Thus, under dark the concentration of ascorbate in the lumen can increase and induce a stronger effect on the Mn-claster."
Response: I have added a note on this possibility.
"L392. ROS was abbreviated on first page, but NPQ was not described at all, as well as VAZ (L417)."
Response: Amended in the entire text.
"In my opinion, the part of the point 6 from L391 to 439 is the weakest in the manuscript. This is written very confusingly with many abbreviations and in total with a little concern to ascorbate involvement. In addition, the author writes about PsbS, but not about Lhcsr3, which is an analogue in green algae. I would recommend to rewrite this part with a strong shortening to one paragraph."
Response: I have reformulated the paragraph, decreased the amount of abbreviations and decreased its length by about 300 words. I have also clarified if the text was about vascular plants or algae.
"In general it seemed to me that the manuscript describes all ascorbate presented in a plant (algae) cell but the title indicate ‘chloroplastic ascorbate’. Can the author explain to me this discrepancy? Should the title be corrected?"
Response: Ascorbate has many other functions that are unrelated to photosynthesis or chloroplastic ascorbate. These are only briefly mentioned in the Introduction, but not reviewed thoroughly. However, I have shortened the title to “The functions of chloroplastic ascorbate in vascular plants and algae”.
Reviewer 2 Report
The review paper is focused on the role of chloroplastic ascorbate in photosynthesis and metabolism in vascular plants and algae.
In my opinion, the manuscript is quite interesting and valuable for plant biologists.
However, I recommend to extend the description of molecular biology aspects involved in AsA biosynthesis and turnover in plant cells (e.g. gene expression studies, next generating sequencing of microRNAs, immunoblots of isoenzymes). In addition, I suggest:
- removing some older citations, and replace them with the new ones,
- adding some schemes and/or diagrams in order to improve the graphical layout for readers,
- minor spell check of English language and style is required.
Author Response
I thank the Reviewer for the positive evaluation of my MS. I have considered the comments and questions and modified the text accordingly.
"The review paper is focused on the role of chloroplastic ascorbate in photosynthesis and metabolism in vascular plants and algae.
In my opinion, the manuscript is quite interesting and valuable for plant biologists.
However, I recommend to extend the description of molecular biology aspects involved in AsA biosynthesis and turnover in plant cells (e.g. gene expression studies, next generating sequencing of microRNAs, immunoblots of isoenzymes)."
Response: In this review I aimed at focusing on the role of chloroplastic ascorbate. The paper is already quite long (10 pages +7 pages of references) that I would not like to extend any further. Besides, a review paper was just published on this matter by Terzaghi and De Tullio (Front. Plant Sci. 13:1096549, doi: 10.3389/fpls.2022.1096549) that I cite in the new version.
"In addition, I suggest:
- removing some older citations, and replace them with the new ones,"
Response: I have added about 10 references from the last few years and removed older, unnecessary ones.
"- adding some schemes and/or diagrams in order to improve the graphical layout for readers,"
Response: I have improved the figure according to the suggestion of Reviewer 1. This figure includes the route of ascorbate biosynthesis, information on its regulation, interactions between the different cell compartments regarding ascorbate biosynthesis, ascorbate transporters and the roles of ascorbate in photosynthesis.
"- minor spell check of English language and style is required."
Response: The text was checked for grammar and clarity.
Round 2
Reviewer 1 Report
I am grateful to the author for the correction of the text and title of the manuscript, as well as for answering all my questions and comments.
Thus, I am sure that the manuscript can be published in the present form.